# Insights into HIV-1 Reverse Transcriptase (RT) Inhibition and Drug Resistance from Thirty Years of Structural Studies

**DOI:** 10.3390/v14051027

**Published:** 2022-05-11

**Authors:** Abhimanyu K. Singh, Kalyan Das

**Affiliations:** 1Laboratory of Virology and Chemotherapy, Rega Institute for Medical Research, KU Leuven, Herestraat 49, 3000 Leuven, Belgium; abhi.singh@kuleuven.be; 2Department of Microbiology, Immunology and Transplantation, KU Leuven, Herestraat 49, 3000 Leuven, Belgium

**Keywords:** HIV-RT, NRTI, NNRTI, drug design, structural biology, crystallography, cryo-EM

## Abstract

The enzyme reverse transcriptase (RT) plays a central role in the life cycle of human immunodeficiency virus (HIV), and RT has been an important drug target. Elucidations of the RT structures trapping and detailing the enzyme at various functional and conformational states by X-ray crystallography have been instrumental for understanding RT activities, inhibition, and drug resistance. The structures have contributed to anti-HIV drug development. Currently, two classes of RT inhibitors are in clinical use. These are nucleoside/nucleotide reverse transcriptase inhibitors (NRTIs) and non-nucleoside reverse transcriptase inhibitors (NNRTIs). However, the error-prone viral replication generates variants that frequently develop resistance to the available drugs, thus warranting a continued effort to seek more effective treatment options. RT also provides multiple additional potential druggable sites. Recently, the use of single-particle cryogenic electron microscopy (cryo-EM) enabled obtaining structures of NNRTI-inhibited HIV-1 RT/dsRNA initiation and RT/dsDNA elongation complexes that were unsuccessful by X-ray crystallography. The cryo-EM platform for the structural study of RT has been established to aid drug design. In this article, we review the roles of structural biology in understanding and targeting HIV RT in the past three decades and the recent structural insights of RT, using cryo-EM.

## 1. Discovery of Reverse Transcriptase and Its HIV Connection

The history of reverse transcriptase (RT) dates back to 1970, when the enzyme was discovered independently by Temin and Baltimore in virions of RNA viruses [1,2]. The enzyme carried an RNA-dependent DNA-polymerase (RDDP) function to synthesize DNA complementing an RNA, the reverse flow of genetic information that was contrary to the central dogma of molecular biology [3]. Thereby, the process was termed reverse transcription, or retrotranscription. The viruses encoding RT for this essential step of converting RNA into DNA as part of their replication cycle were called retroviruses. Further investigations attributed RT with additional capabilities of DNA-dependent DNA-polymerase (DDDP) activity and degrading RNA of an RNA/DNA duplex via its Ribonuclease H (RNase H) activity [4,5,6]. A decade later, retrovirus and RT were about to become household names (Figure 1).

The early 1980s witnessed a rise in cases of a new disease with unusual symptoms linked to immune system deficiencies. In 1981, rare lung infection was observed in a group of patients who experienced severe non-functional immune disorder [7]. While the underlying cause or the causative agent was not known, the condition was termed acquired immunodeficiency syndrome (AIDS) in 1982 by the Center for Disease Control (CDC). The detection of reverse transcription activity in lymphocyte cells taken from AIDS patients identified the presence of a retrovirus [8,9], which was subsequently named human immunodeficiency virus (HIV)—the etiological agent of AIDS. As the devastating AIDS pandemic spread through the world, the scientific community rushed to find treatment options against the disease, bringing enzymes exclusive to the virus into the forefront of drug research targets, and provided a thrust to broader retroviral research programs. HIV RT, being one of the most crucial enzymes involved in viral replication, was perceived as a prime target. A number of effective RT inhibitors have been developed for clinical use against HIV infection (Figure 2).

## 2. HIV Replication Cycle and RT

The HIV replication cycle starts with a mature virion attachment via its surface glycoproteins (Env; gp120 and gp41 subunits) to host-cell receptors CD4 and CCR5/CXCR4. The interactions trigger the fusion of the viral and host-cell membranes facilitating the release of the viral core into the host-cell cytoplasm by endocytosis [10,11,12]. A fullerene cone-shaped HIV capsid core contains essential components of the virus, including three viral enzymes, RT, integrase (IN), and protease (PR); and two copies of the single-stranded viral RNA (ssRNA) genome of 9.7 kb size [13,14]; additionally, the host-cell components, such as cyclophilin A and tRNA^Lys3^, are also packed. The viral core undergoes partial or full uncoating inside the cell cytoplasm via a mechanism that is not well understood [15,16,17,18]. In the cytoplasm of an infected cell, HIV replication starts with the RT activity that converts the viral ssRNA genome to a double-stranded DNA (dsDNA). The synthesized viral DNA is transported to the nucleus and integrated into the host-cell chromosome by the viral enzyme integrase, which is a hallmark of all retroviruses, including HIV. Subsequently, the infected cell machinery is engaged in producing the viral components, assembling the components as premature virions, and budding/releasing the premature viruses, which undergo maturation by the enzyme protease processing the viral gag and gag–pol polyproteins. [19].

The enzyme RT binds a double-stranded nucleic acid substrate and catalytically synthesizes DNA. RT accomplishes the viral DNA synthesis in multiple steps that include the enzymatic activities, RDDP, RNase H cleavage, and DDDP [20]. At the 5′-end of the ssRNA viral genome, a stretch of 18-nucleotides known as the primer binding site (PBS) is complementary to the 3′-end 18-nucleotides segment of human tRNA^Lys3^. Both hybridize to form an 18-mer dsRNA duplex that is recognized by RT to initiate the first (−) strand DNA synthesis from the 3′-end of the tRNA^Lys3^, complementing the viral ssRNA genome as the template. RDDP continues till the 5′-end of the viral ssRNA, generating an RNA/DNA hybrid which is recognized for selective degradation of RNA by the RNase H function of RT. The RNA cleavage makes the free nascent (−) strand DNA available to hybridize with the viral ssRNA, as the 5′ and 3′ ends of the viral RNA genome share identical sequence repeats (termed as R). The process is termed the first strand transfer [21]. During the cleavage of RNA by the RNase H from the RNA/DNA hybrid, regions rich in purine sequence (polypurine tract or PPT) located near the 3′-end of the viral RNA are not cleaved. RT starts (+) strand DNA synthesis from the PPT sites by using those as primers [22,23]. The (+) strand DNA synthesis proceeds till the first 18-nucleotides of the annealed tRNA, where the presence of a modified base blocks RT, generating a (+) strand strong-stop DNA. Subsequently, the RNA segment of the tRNA/DNA hybrid is removed by RNase H activity, freeing the PBS sequence of the (+) strand DNA and allowing it to anneal to the complementary site near the 3′-end of the extended (−) strand DNA [24], a process termed as the second strand transfer. Bidirectional DNA synthesis continues, resulting in the synthesis of a viral dsDNA with long terminal repeats (LTRs) at both ends, and the dsDNA is translocated to the cell nucleus for integration with the host genome (for an in-depth description of retroviral replication, see Reference [25]).

## 3. Structures of RT

HIV-1 RT is derived from the Pol polyprotein as 560 amino acids long p66 that form homodimers in which one p66 is in the enzymatically active form [26], and the second copy (p66’) is rearranged to provide a structural foundation. The RNase H domain at the C-terminal of p66’ is disordered and subsequently cleaved by the HIV protease to form the functional RT as a p66/p51 heterodimer; p51 has about 440 amino acids residues [27,28,29]. Crystal structures of RT in complex with NNRTI nevirapine (NVP), and RT with a double-stranded DNA were the first to be solved [30,31,32], and the structures highlighted the key features of this multifunctional enzyme. Similar to all other polymerases, the polymerase domain of RT has a hand-like conformation [33] with fingers (1–85 and 118–155), palm (86–117 and 156–236), thumb (237–318), and connection (319–426) subdomains. The inactive p51 subunit has the same polymerase subdomains as in p66; however, a different spatial arrangement of p51 results in an asymmetric organization as the heterodimer (Figure 3A,B). RT engages template/primer in a cleft extending from the polymerase active site, containing the catalytic aspartates (D110, D185, and D186), to the RNase H active site that are separated by about 18-nucleotide duplex length. (Figure 3A). 

The structure of a catalytic HIV-1 RT/DNA/dTTP ternary complex was determined by using a disulfide crosslinking of RT thumb residue Q258C with a modified nucleotide base in the dsDNA minor groove to stabilize the complex [34]. The dTTP-binding in the ternary complex is accompanied by the fingers closing to form a dNTP-binding pocket and chelation of two Mg^2+^ ions involving the triphosphate moiety of dTTP and the catalytic aspartates (Figure 3C,D) [35]. Subsequently, multiple structures were determined by trapping various conformational, functional, and inhibited states of wild-type and drug-resistant HIV-1 RTs. The structures and biophysical studies reveal that RT is highly flexible, and this dynamic enzyme undertakes spatial rearrangements for carrying out its functions [36,37].

## 4. Drugs Targeting RT

Two main classes of RT drugs have been developed as anti-HIV medications. Those are (a) nucleoside/nucleotide RT inhibitors (NRTIs) and (b) non-nucleoside RT inhibitors (NNRTIs) (Figure 2). NRTIs and NNRTIs are widely included in HIV treatment regimens and have greatly helped in turning AIDS into a manageable condition. However, due to the low fidelity of RT and high rate of viral replication, HIV drugs face challenges of drug resistance, which continues to be a major problem. HIV treatment is life-long; therefore, long-term use of a drug can lead to toxicity. The periodic addition of new drugs to the existing pool of HIV drugs helps overcome the drug resistance and toxicity. Apart from two current classes of RT drugs, NRTI and NNRTIs, additional druggable sites exist in RT. The following sections outline a structural perspective on the existing RT drugs, the challenges of drug resistance, and other classes of RT inhibitors and druggable sites.

### 4.1. Nucleoside/Nucleotide RT Inhibitors (NRTIs)

The NRTIs were the first class of inhibitors discovered as anti-HIV medication. Zidovudine (AZT) was the first HIV drug approved in March 1987, and to date, eight NRTIs have been approved as HIV drugs (Figure 2A). These are Zidovudine (ZDV/AZT, Retrovir), Didanosine (ddI, Videx), Zalcitabine (ddC, Hivid), Stavudine (d4T, Zerit), Lamivudine (3TC, Epivir), Abacavir (ABC, Ziagen), Tenofovir (TFV, Viread), and Emtricitabine (FTC, Emtriva). After being delivered to an infected cell, an NRTI must be converted to a dNTP analog by cellular kinases [38]. Both entry and phosphorylation of NRTIs are the key limitations in developing effective NRTI drugs. In contrast to nucleoside analogs that require three phosphorylation steps, nucleotide analogs have the advantage of not requiring the addition of α-phosphate. Most nucleoside analogs, such as AZT and 3TC, enter the cell by diffusing through the cell membrane. In contrast, the nucleotide analogs—such as TFV, which has a phosphonate group—have poor cellular uptake and are formulated as prodrugs [39]. 

The phosphorylated NRTIs bind RT as dNTP analogs and catalytically incorporated into the growing DNA chain by RT; the pyrophosphate moiety is removed as the reaction byproduct. In general, an NRTI has modified sugar moiety lacking the 3′-OH group. Therefore, once incorporated, an NRTI acts as a chain terminator and blocks the growth of the DNA strand. New NRTIs are being developed that function as delayed chain terminators or translocation-deficient inhibitors [40,41], or they inhibit by other indirect mechanisms [42]. The nucleotide analogs are also highly effective in inhibiting RNA-dependent RNA polymerases (RdRps) of RNA viruses [43]. For example, Sofosbuvir containing regimen successfully cures hepatitis C (HCV) infection [44], and remdesivir is a broadly effective antiviral that has been approved to treat SARS-CoV2 [45]. Even though new studies show reduced efficacy of remdesivir against SARS-CoV-2 [46], other nucleoside analogs, such as molnupiravir (MK-4482) [47], are expected to perform better, and molnupiravir is in the pipeline for FDA approval. The knowledge gained from NRTI studies has been significantly contributing to the development of nucleoside analogs against viral RdRps.

In common with most HIV drugs, clinical resistance mutations emerge in response to NRTI drugs. NRTI resistance is usually conferred by the emergence of RT mutations that are generated as the consequences of the errors introduced in the replication process due to the low fidelity of RT and the high rate of HIV replication [48]. The dNTP binding site and surrounding residues are mutated to confer NRTI resistance (Figure 4A) by discriminating an NRTI-TP from dNTP at the binding or incorporation stage [49], or by the removal of an incorporated NRTI from the 3′-end of DNA primer [50]. Pathways leading to resistance mutation accumulation are complex, and over the years, biochemical and structural studies have made significant contributions to their understanding [51,52]. Two primary mechanisms behind NRTI resistance have been established: (a) resistance by excision and (b) resistance by exclusion [53].

A set of mutations, namely M41L, D67N, K70R, L210W, T215Y, and K219Q, emerges in patients under AZT treatment [54,55,56]. This set of mutations, termed as thymidine analog mutations (TEMs), was later shown biochemically to help excise AZT after its incorporation into the DNA primer strand [57,58]. Thymidine analogs such as d4T and AZT are particularly vulnerable to such excision via a process known as pyrophosphorolysis (Figure 4B), which essentially is the reverse of DNA polymerization [59]. The crystal structure of RT/DNA/AZTppppA and related complexes revealed the structural basis of AZT excision mediated by ATP and highlighted the crucial roles that K70R and T215Y mutations play [60,61,62]; AZTppppA is the product of ATP-mediated AZT excision. The K70R and T215Y mutations adjacent to the dNTP binding pocket (Figure 4C) help create a sub-pocket to accommodate ATP, which functions as the pyrophosphate donor to facilitate the pyrophosphorolysis reaction of excision [60]. Phenotypic and biochemical studies have revealed more complex patterns promoting NRTI excision that include a dipeptide insertion in the β3–β4 fingers loop to help excise other NRTIs [60,63,64,65]. 

Resistance to NRTI can also result from certain mutations primarily occurring in the dNTP binding pocket and conserved YMDD motif of RT that contains the catalytic residues D185 and D186; the YXDD motif is conserved in all RDDPs. Several mutations have been characterized which allow RT to discriminate between an NRTI-TP and a natural dNTP (reviewed in References [48,51,52,66]). K65R mutation is selected for resistance to TFV, ABC, ddI, and d4T [67,68,69,70]. The structure of K65R mutant RT in a complex with TFV or dATP showed a stacking interaction between K65R and R72 side chains, forming a structurally rigid platform that can discriminate TFV from dATP [71]. The structural basis of lamivudine (3TC) resistance was understood with the help of the crystal structures of binary and ternary complexes of wild-type and M184I mutant RT [34,72,73]. The only non-conserved position M184 in the YXDD motif mutates to confer resistance to NRTIs containing L-deoxyribose ring. The switched L-deoxyribose ring of 3TC/FTC is discriminated by the β-branched rigid side chain of the mutated M184I/V residue at the polymerase active site. The hepatitis B polymerase also uses a similar mechanism to discriminate 3TC/FTC from dCTP [74,75].

The NRTIs can work in synergy among themselves and with other HIV drugs [76]. For instance, TFV and 3TC/FTC work in combination [77], whereas the thymidine analogs AZT and d4T are not effective in combination [78]. Experimental NRTIs, such as Apricitabine (ATC) and Elvucitabine (L-d4TC), that are currently under clinical development show better toxicity profiles than some approved NRTIs [79,80,81]. Similarly, the pursuit of improved NRTIs led to the discovery of 4′-ethynyl-2-fluoro-2′-deoxyadenosine (EFdA) as an RT inhibitor that exerts its effect by interfering with RT translocation over its substrate [82]. EFdA has 4′-ethynyl and 2-fluoro substitutions on its sugar and base moieties, respectively. EFdA demonstrates excellent selectivity and stability profiles, as well as potency against drug-resistant HIV mutants [83,84,85,86]. A poor affinity for DNA polymerase γ suggests the minimal mitochondrial toxicity of EFdA [87]. The crystal structure of EFdA in complex with RT/DNA has shown that the inhibition by EFdA is mediated by the 4′-ethynyl substitution in the sugar ring, which interacts with a conserved hydrophobic sub-pocket in the polymerase active site [41]. In summary, the structure and biochemical studies have helped understand inhibition and drug-resistance mechanisms and provide new opportunities to discover novel NRTIs. 

### 4.2. Non-Nucleoside RT Inhibitors (NNRTIs)

Nevirapine and tetrahydroimidazo [4,5,1-*jk*][1,4]benzodiazepine-2(1*H*)-thione and -one (TIBO) derivatives were the first NNRTIs to be discovered and provided the foundation for discovery of new NNRTIs [88,89]. The first structure of HIV-1 RT was determined in a complex with nevirapine [31]. The NNRTIs are specific to HIV-1 RT and are not effective inhibitors of HIV-2 RT. The NNRTI-binding pocket (NNIBP) is constituted by several hydrophobic residues (Figure 2 and Figure 5A) and is located ~10 Å away from the polymerase active site [31,90]. All NNRTIs act as allosteric inhibitors of RT, and their binding induces structural changes to the polymerase active site and interferes with viral DNA synthesis. Because of high selectivity, NNRTIs do not directly interfere with cellular polymerases and exhibit significantly low cellular toxicity when compared to NRTIs. The potential source of NNRTI toxicity could be by promiscuous binding to interfere with unknown cellular functions.

The comparison of RT structures with and without an NNRTI [32,34,91] reveals that the NNIBP is formed upon an inhibitor binding and the pocket is highly adaptable when compared to a typical substrate-binding pocket. Thereby, NNIBP provides wide opportunities for NNRTI design. On the contrary, the NNIBP region is not required to be conserved for RT to carry out its function; therefore, pocket mutations can readily emerge to confer NNRTI resistance without significantly compromising viral fitness. For example, a single dose of nevirapine selects for the Y181C mutation in patients [92,93,94]. Most of the pocket residues can mutate to confer NNRTI resistance via steric hindrance (L100I or G190A/S), and/or loss of key protein–inhibitor interactions (V106A, V179D, Y181C, Y188L, and F227C/L). Biochemically, the pocket entrance mutations, such as K101E/P, K103N, and E138K, can affect the kinetics of NNRTI binding [95,96]. In summary, the NNRTI design provides unique opportunities and challenges. Six NNRTI drugs have been approved for treating HIV-1 infections—nevirapine (NEV, Viramune), delavirdine (DLV, Rescriptor), efavirenz (EFV, Sustiva), etravirine (ETR, Intelence), rilpivirine (RPV, Edurant), and doravirine (DOR, Pifeltro) (Figure 2B); doravirine is the latest FDA-approved RT-inhibiting HIV-1 drug. 

Following the structure solution of the RT/NVP complex, multiple wild-type and mutant RT/NNRTI complex structures were determined [97,98,99,100,101], and the structural information was used in designing new NNRTIs [102]. The design and discovery of the diarylpyrimidine (DAPY) drugs ETR and RPV proved to be very effective, because they could overcome the impacts of common NNRTI-resistance mutations. A new concept of using conformational flexibility in designing drugs to overcome the impact of drug-resistance mutations emerged from the discovery of DAPY class of inhibitors. The use of torsional flexibility can help a drug to reorient and reposition to retain efficacy even in the face of pocket mutations (Figure 5A,B) [103,104]. For example, the second-generation NNRTIs RPV and DOR have such torsional flexibilities. E138K + M184I/V mutations occur in patients treated with RPV and the mutations cause low-level resistance, whereas no noticeable clinical resistance mutation emerges in response to DOR. DOR and RPV have complementary efficacies despite both being NNRTIs [105].

The NNRTI-inhibited RT/nucleic acid polymerase complexes provide the snapshots that an NNRTI displaces the primer 3′-end of a dsDNA or an RNA/DNA substrate away from the polymerase active site. The NNRTI-induced changes in the dNTP-binding site reveal that an NNRTI also impacts the binding of dNTP substrates [106]. Later, it was found that the NNRTI-bound RT conformation resembles the conformation of the RT/dsRNA initiation complex [107,108]. The difference in the binding mode of dsRNA compared to an elongation substrate explains the less efficient nucleotide incorporation in the RT initiation state [109]. Recent high-resolution cryo-EM structures of NNRTI-inhibited RT/dsRNA and biochemical studies showed that NNRTIs also inhibit RT initiation [110]. The RT conformations in NVP- and EFV-inhibited RT/dsRNA/NNRTI, RT/RNA-DNA/NNRTI, and RT/dsDNA/NNRTI superimpose well (Figure 5C) suggesting that NNRTIs trap RT in a common conformational state. However, the recent cryo-EM structures of new-generation NNRTIs RPV and DOR in complexes with RT/dsDNA show key differences near polymerase active site and in the track of dsDNA when compared to earlier NNRTIs (Figure 5C,D) [111]. Impacts on the RT/DNA catalytic complex, which differs for different NNRTIs, provide important insights for understanding NNRTI resistance mutations. Therefore, NNRTI-inhibited RT/nucleic acid structures may be valuable in designing new NNRTI, and the new cryo-EM platform developed for the rapid structure determination of RT/DNA/NNRTI complexes will be useful in NNRTI design.

## 5. Other Targets of RT and Future Perspective

Apart from the NRTIs and NNRTIs that are successful drugs, there exist additional functions and/or sites of RT as potential drug targets. Among those, the RNase H activities, nucleoside-competing RT inhibitors (NcRTIs), and recently discovered transient P-pocket are attractive for drug design.

### 5.1. RNase H

The RNase H is the second active site that catalytically cleaves the RNA strand from an RNA/DNA duplex, and, thereby, the RNase H activity has been considered as an important target for developing anti-HIV drugs. The structure of the HIV-1 RNase H domain only was reported [112] even before the complete structure of RT. The structure of RNA (PPT sequence)/DNA in complex with HIV-1 RT showed that the RNA is positioned away from the RNase H active site, as the PPT sequence is a poor substrate for RNase H cleavage [113]. Subsequent studies [114,115,116] have established the structural basis for the RNase H cleavage activity by HIV-1 RT. 

RNase H shares the common architecture of an endonuclease active site that also exists in successfully targeted active sites of HIV-1 integrase and influenza cap-snatching endonuclease [117,118,119]. The inhibitors, such as diketo acid derivatives, share a common moiety that chelates the active-site cations in all three enzymes [120]. Structures of RNase H inhibitors in complexes with HIV-1 RT have been determined with the aim to develop RNase H inhibitors as HIV drug candidates (Figure 6A) [121,122,123]. All inhibitors share an analogous mode of active-site metal chelation; however, no significant additional inhibitor–protein interactions have been observed for any of the inhibitors. In contrast to the HIV-1 integrase and influenza cap-snatching endonuclease active sites that form well-defined pockets to accommodate inhibitors, the RNase H active site is rather a flat cleft, which does not permit favorable interactions for improving the binding affinity for known RNase H inhibitors beyond the metal chelation. In summary, even though RNase H activity is an attractive drug target, the attempts to develop drug candidates have not yet been highly effective in part due to the flat shape of the active site.

### 5.2. Nucleoside-Competing RT Inhibitors (NcRTIs)

In contrast to NRTIs that are DNA chain terminators, NcRTIs directly compete with dNTP binding. There are two classes of NcRTIs—metal-independent and metal-chelating. INDOPY-1 (5-methyl-1-(4-nitrophenyl)-2-oxopyrido [3,2-b]indole-3-carbonitrile) was discovered as a dNTP-competitive inhibitor [124] that showed ~15x reduced susceptibility to RT containing M184V and Y115F mutations, and enhanced susceptibility to RT containing TFV-resistance mutation K65R [125]. The crystal structure of the HIV-1 RT/DNA/INDOPY-1 complex revealed that the inhibitor binds at the polymerase active site primarily by hydrophobic stacking with the first base-pair of the dsDNA substrate [126]. The INDOPY-1 occupies the position of the base and sugar of a dNTP and the base of the first templet overhang. (Figure 6B) However, INDOPY-1 is not involved in metal chelation.

The second class of NcRTI is the polymerase active-site metal chelator represented by the α-carboxy nucleoside phosphonate (α-CNP) class of compounds [127]. An α-CNP binds RT/DNA catalytic complex at the polymerase active site, mimicking a dNTP substrate (Figure 6C). The inhibitor base-pairs with the template overhang and, importantly, chelates both catalytic Mg^2+^ ions (Figure 6D). An α-CNP has three chelations that mimic the chelation of a dNTP at the active site (Figure 6E), and, thereby, the binding of α-CNP completes the octahedral coordination environment for both catalytic Mg^2+^ ions at the active site. The metal chelating environment at the active sites of RNA and DNA polymerases is highly conserved. Therefore, the discovery of chemical moieties that can effectively replace the chelation of the triphosphate of a dNTP (or NTP) at the polymerase active site has broader implications for developing NcRTIs targeting different viral RNA or DNA polymerases. Analogous to the development of HIV integrase inhibitors and influenza endonuclease inhibitors from a common metal-chelating moiety, a polymerase active-site-chelating moiety can be elaborated to develop different polymerase-specific inhibitors.

### 5.3. P-Pocket

For facilitating catalytic steps, an enzyme exists in transient states which are short-lived but crucial for the successful completion of the enzymatic reaction [128]. These states, in general, have been considered as important targets for drug design [129]. RT is highly dynamic and acquires different conformational states to perform its multiple functions [37,130,131,132,133,134,135]. During the polymerization process, RT slides over a nucleic acid substrate, as shown by single-molecule fluorescence resonance energy transfer studies [36,136], which indicate the existence of transient states. A transient state of RT was trapped and stabilized by crystal contacts [137]. The crystal structure showed RT slides ahead of dsDNA substrate such that the primer 3′ terminus occupies the P-1 site (P-1 complex), thus creating a transient P-pocket (Figure 6F). The P-pocket was probed for small molecule binding by crystal-based fragment screening, using a fragment library of about three hundred compounds. Two fragments were identified binding the P-pocket. However, the crystals of the transient state complex were not very stable for the soaking of small molecules in the follow-up optimization attempts; the crystals were primarily damaged or losing diffraction resolution upon soaking. Therefore, the P-1 complex was engineered to be stable in solution using a modified DNA aptamer, and the structures of compounds bound at the P-pocket of the engineered P-1 complex were determined by single-particle cryo-EM [137]. The findings provide a new target site and experimental platform to effectively use cryo-EM for optimizing compounds against P-pocket. The structural motifs—primer grip, polymerase active site, DNA duplex, and the template overhang—are less susceptible to mutations that form in the walls of the P-pocket. Effective structure-based drug design effort can aid the discovery of potent P-pocket inhibitors. Because of the structural conservation and function importance of the region, it may be difficult for a virus to develop resistance to such a P-pocket inhibitor without a fitness cost. The pocket region is conserved in HIV-1 subtypes (or clades) and in HIV-2 RT. Therefore, a P-pocket inhibitor is likely to have broad effectiveness across the HIV-1 and HIV-2 species. Additionally, the existence of analogous pockets in other viral polymerases is very likely, and, thereby, the concept of targeting the P-pocket may be extended to find and target such transient pockets in other viral RNA and DNA polymerases.

The recent emergence of pan-resistance in HIV patients has led to treatment failure in all five classes of HIV drugs—NRTIs, NNRTIs, protease inhibitors (PIs), integrase strand-transfer inhibitors (INSTIs), and fusion inhibitors [138]. New treatment strategies and new drugs are to be developed to deal with pan resistance. The use of long-acting and slow-release formulations of drugs such as CABENUVA, a combination of cabotegravir and RPV, [139] may help limit the emergence of multidrug-resistant strains by improving treatment adherence and preventing drug holidays. A study demonstrates that an optimized antiviral treatment strategy based on accurate identification of resistance mutations and the use of an anti-CD4 monoclonal antibody ibalizumab is successful in suppressing pan-resistance [140]. While the result is encouraging, an antibody-based treatment is not affordable or logistically feasible in most parts of the world. Continuing multidisciplinary research is essential to meet the emerging challenges in HIV drug resistance and treatments.

## Figures and Tables

**Figure 1 viruses-14-01027-f001:**
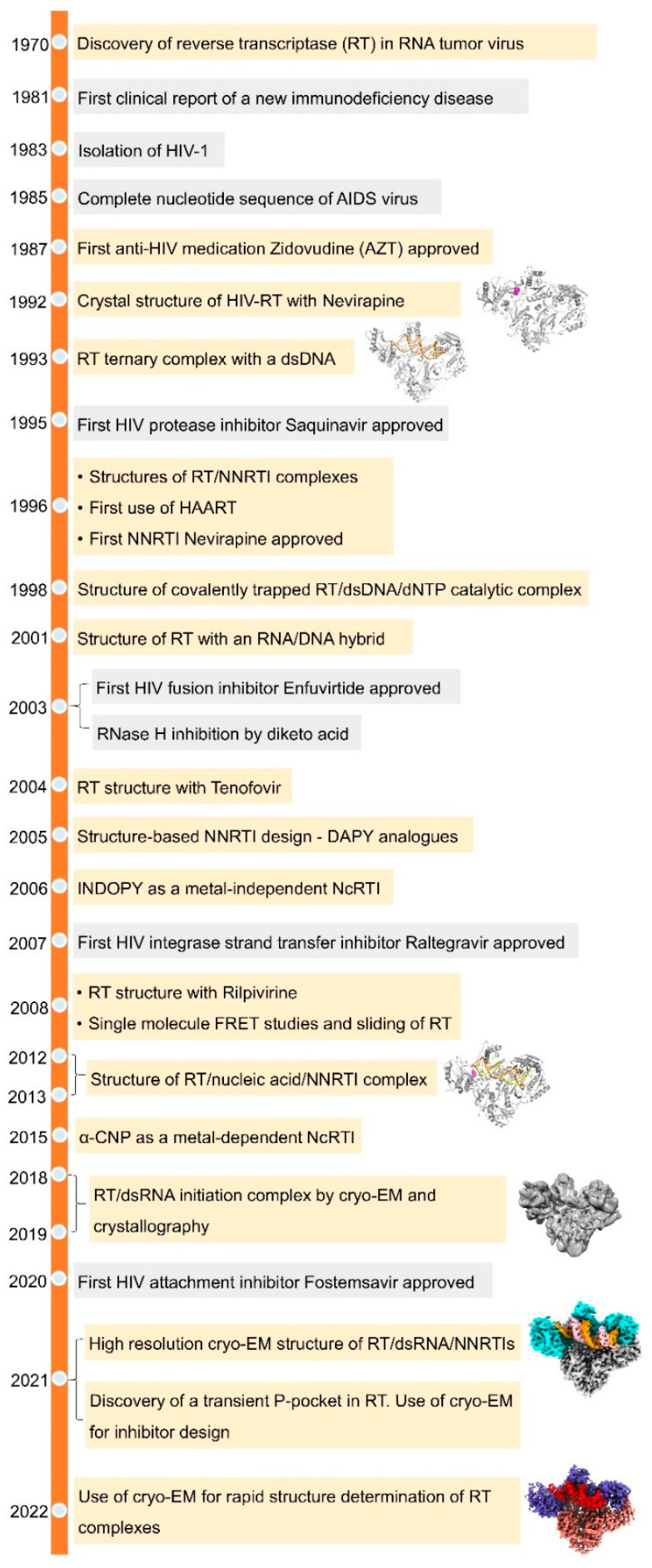
**A historical perspective of key developments in HIV-1 research.** Events concerned with HIV-1 RT and its structural biology are highlighted in cream color.

**Figure 2 viruses-14-01027-f002:**
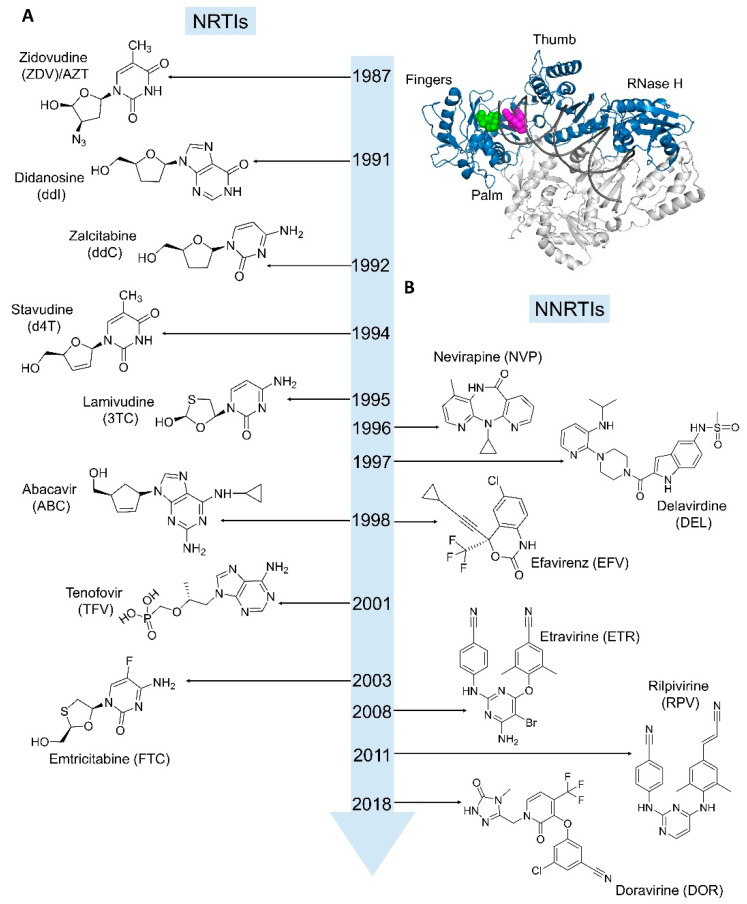
**Timeline of HIV-1 RT-inhibiting drugs approved for clinical use.** (**A**) The nucleoside/nucleotide analogs (NRTIs) act as DNA chain terminators. (**B**) NNRTIs bind to a pocket away from the polymerase active site and allosterically inhibit RT. Binding sites for dNTP/NRTI-TP (green) and NNRTI (magenta) in the p66 subunit (blue) of RT are shown on the top right panel. The p51 subunit is in gray.

**Figure 3 viruses-14-01027-f003:**
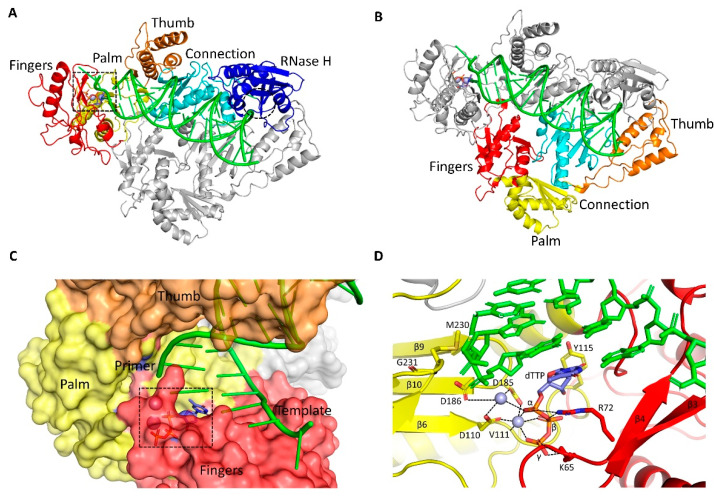
**Binding of dNTP to HIV-1 RT.** (**A**) Structure of RT in complex with a dsDNA, and dTTP bound at the polymerase active site (black dotted box; PDB ID = 1RTD [34]). The dsDNA is in green, dTTP as the blue stick model, and RNase H active site is highlighted as the dotted circle. The p66 (red, fingers; yellow, palm; brown, thumb; cyan, connection; and blue, RNase H) and p51 (gray) subunits constitute the hetero-dimeric RT. The template/primer in a cleft extends from the polymerase active site to the RNase H active site. (**B**) Different spatial arrangement of the subdomains in p51 (red, fingers; yellow, palm; brown, thumb; and cyan, connection) as compared to that in p66 (**A**); p66 is in gray. (**C**) A surface view of the dNTP-binding pocket highlighting the key p66 subunits involved in the pocket formation. The bound dTTP in the pocket is highlighted by a black dotted box. (**D**) A zoomed view of the dNTP-binding site showing metal-chelation and other interactions of the bound dTTP. The dNTP-binding site is composed of residues from palm and fingers; some are conserved, and some mutate to confer NRTI resistance [34].

**Figure 4 viruses-14-01027-f004:**
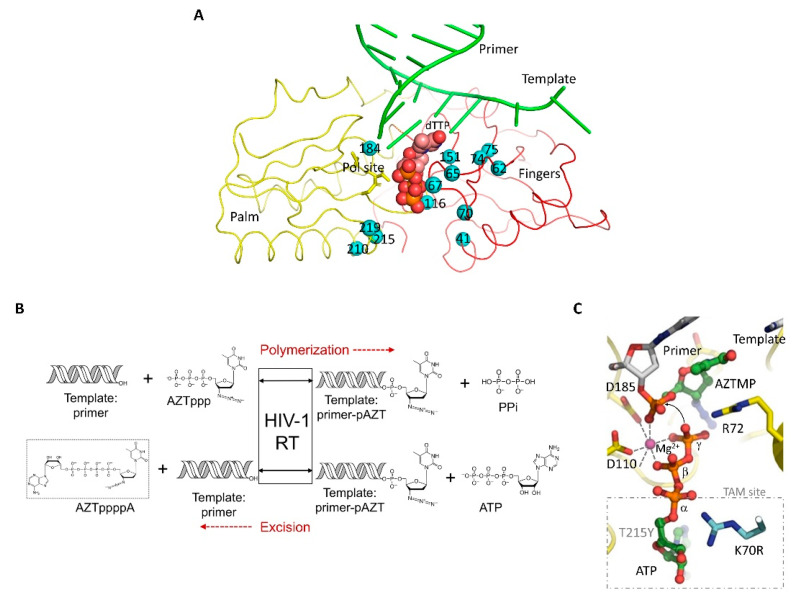
**NRTI resistance and pyrophosphorolysis.** (**A**) Sites for commonly observed NRTI-resistance mutations (cyan spheres) are located around the dNTP-binding site. (**B**) A schematic representation showing DNA polymerization (top) and AZT excision (bottom). (**C**) Mutations adjacent to the dNTP binding pocket of RT help accommodate ATP, which acts as the pyrophosphate donor in the excision reaction. Panels B and C are reprinted with permission from Reference [61]; copyright (2010) Nature Publishing Group.

**Figure 5 viruses-14-01027-f005:**
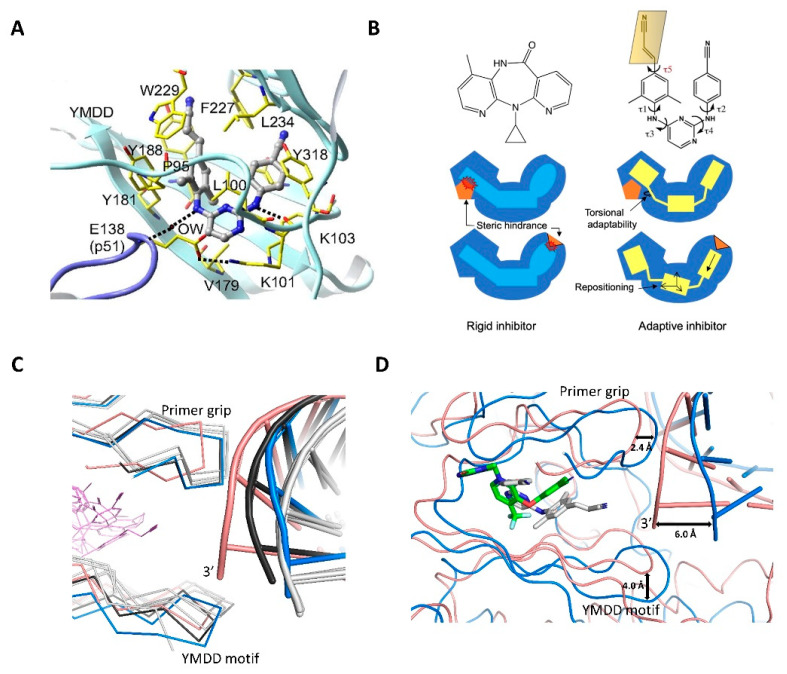
**NNRTI binding and its impacts on RT and nucleic acid substrates.** (**A**) Binding of RPV (gray stick) to the NNRTI-binding pocket (NNIBP) (PDB ID = 2ZD1); key residues in NNIBP are highlighted in yellow; the figure is reprinted from Reference [103]; copyright (2008) National Academy of Sciences. (**B**) Schematic comparison of a rigid (left) and a flexible inhibitor (right) highlighting the advantage of strategic flexibility to overcome the impact of resistance by adapting to the pocket changes [104]; τ represents rotatable bonds. (**C**) Overlay of wild-type RT/DNA/DOR (salmon; PDB ID = 7Z2G), RT/DNA/RPV (blue; PDB ID = 7Z2D), RT/DNA/NVP (black; PDB ID = 7Z24), RT/dsRNA (PDB ID = 7KJV), RT/dsRNA/NVP (PDB ID = 7KJX), and RT/dsRNA/EFV (PDB ID = 7KJW); all RT/dsRNA structures are in gray. The comparison shows that NNRTIs trap RT in a common conformational state, except for DOR and RPV complexes; the primer grip and the nucleic acid track in the DOR complex, and the YMDD motif in the RPV complex deviate the most. (**D**) Cα-superimposition of RT/DNA/DOR (salmon) and RT/DNA/RPV (blue) structures. The inhibitors (DOR in green; RPV in gray) occupy the same pocket space; however, significant shifts in primer 3′-end, conserved YMDD loop, and the primer grip are observed between two structures [111].

**Figure 6 viruses-14-01027-f006:**
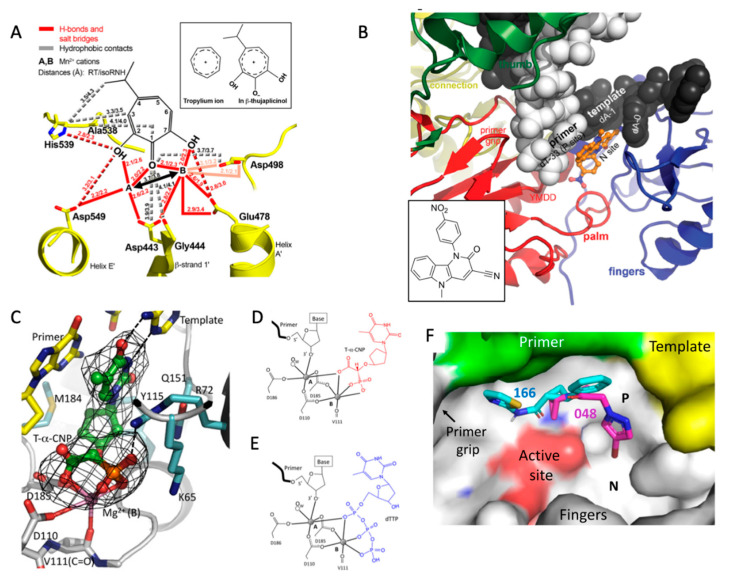
**Other key potential target sites of HIV-1 RT.** (**A**) Binding of β-thujaplicinol at the RNase H active site via chelation to Mg^2+^ ions A and B; reprinted with permission from Reference [121]; copyright (2009) Elsevier Ltd. (**B**) Binding of the NcRTI, INDOPY-1 at the polymerase active site (PDB ID = 6O9E); the figure is reprinted with permission from Reference [126]; copyright (2019) American Chemical Society (**C**) Binding of α-CNP at the polymerase active site of HIV-1 RT (PDB ID = 4R5P), and the polymerase active site metal chelation of α-CNP (**D**) and dTTP (**E**); the figures (**C**–**E**) are reprinted from Reference [127]; copyright (2015) National Academy of Sciences. (**F**) Binding of two drug-like fragments 166 and 048 to the transient P-pocket (PDB IDs: 7OZ5 and 7OXQ, respectively).

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
