# Peer review of "Insights into HIV-1 Reverse Transcriptase (RT) Inhibition and Drug Resistance from Thirty Years of Structural Studies"

_viruses, 2022, doi:10.3390/v14051027_

Round 1

Reviewer 1 Report

The discovery of reverse transcriptase was a major biological breakthrough demonstrating that genetic information could flow from RNA to DNA rather than the reverse. After the identification of HIV as a retrovirus, the HIV-1 RT became a major target for drug development and, as noted in the review, was the first HIV enzyme to be successfully targeted by a drug, AZT, originally developed as an anti-cancer agent.  This review was prepared by members of the Arnold group who have been the dominant contributors to providing structural insights into the mechanism, the basis of drug inhibition, and the basis of mutation-dependent drug resistance. Overall, they are largely responsible for our current understanding of HIV RT and of the entire class of reverse transcriptase enzymes.

            My comments deal mostly with minor points, including things that I felt might be useful for inclusion or improve the clarity of the presentation.   

p. 3, Section 1. I found the following sentence not well constructed: "As the devastating AIDS pandemic spread through the world, the scientific community rushed to find treatment options against the disease, thrusting retroviral research in general, and bringing enzymes exclusive to the virus into the forefront as leading drug targets." The pause after the word general ends the phrase without a clear object.  Sentence should be rewritten, e.g.: As the devastating AIDS pandemic spread through the world, the scientific community rushed to find treatment options against the disease, thrusting research into retroviral enzymes and other enzymes exclusive to the virus to the forefront of drug research targets.  

p.4, line 2. In the phrase: "to a host cell receptors" delete the "a" 

p. 3, Figure 2.  I suggest that at the upper right hand side of figure 2, it might be useful to use a different color to indicate the p51 subunit in order to more clearly illustrate the positions of the inhibitors relative to the two subunits.  

p. 4. Section 2. The first paragraph, mentions the viral proteins in the virion. However, in addition to the viral proteins, the virion contains host cell proteins including cyclophilin A, which interacts with the viral capsid structure, and tRNALys3 which, as described in the following paragraph, hybridizes with the viral PBS. I think that the presence of both of these host proteins in the virion also should be noted in the first paragraph.  

p. 4, first paragraph last sentence. According to multiple references, viral maturation involves processing of the entire gag-pro polyprotein, not just the gag polyprotein; in particular, the crucial protease is encoded in pol and not gag. 

p. 4, paragraph 2, line 3, I don't understand the mention of "RDRP", which is presumably RNA-dependent RNA polymerase? I don't think that RT catalyzes RNA polymerization (except perhaps for an occasional error). Later in the paragraph, the authors correctly mention RDDP – RNA-dependent DNA polymerization, which is of course important. Perhaps I have misinterpreted the RDRP? 

p. 4 Section 3, line 4, change disorder to disordered 

p. 5 Figure 3a.  The last sentence on p. 4 denotes the different spatial arrangement of the polymerase subdomains in p51 relative to p66 and cites Figure 3A, but the p51 subdomain arrangement is not apparent in Figure 3A.  Perhaps the authors need to divide 3A into two figures with the subdomains of p66 and p51 color-coded in each.   

p. 5, Figure 3B.  In Figure 3B, Asp186 does not appear to interact with the catalytic Mg ions, which appears to contradict Figure 5D. What is the reason for this apparent contradiction? Perhaps this figure should be replaced with one that is consistent with Figure 5D? I also had some difficulty distinguishing the red and salmon pink fingers and dTTP - mostly because they are close in space and color.  Might be better to use a different color for one of. them.

Paragraph 1, line 2, replace "so far" with "to date" or "as of this review" 

p. 6, Section 5, first paragraph, last sentence: I would qualify the statement: "However, nucleotide analogs have poor cellular uptake and are formulated as prodrugs [39]." – Nucleosides generally exhibit poor cellular uptake and are formulated as prodrugs, however, some of the more hydrophobic nucleosides such as AZT are more able to diffuse through the cell membrane, (Zimmerman et al., 1987; Kong et al., 1992). Perhaps this contributed to the early successful identification of AZT? 

p. 6, second paragraph.  Discussion of the RDRPs, while interesting, is a bit off the mark. I would have rather seen a discussion of the structural work reported by Tu et al. (reference 61) illustrating how mutations facilitate unblocking of site terminators.

p. 6, second paragraph, remove coma from "Even though, new studies..." I suggest authors check the full manuscript for unnecessary comas.

p. 6, Section 5, paragraph 3: Sentence beginning: "Like for all HIV drugs" is ungrammatical. Replace with something like: "In common with most other RT drugs, the efficacy of NRTIs is limited by resistance mutations."  

p. 7. Section 5, paragraph 5: Correct first sentence to read: Resistance to NRTIs can also result from certain mutations primarily occurring in the dNTP binding pocket and highly conserved YMDD loop of RT.  (Not sure it makes sense to mention mutations in a conserved sequence – implies it is not conserved?) 

p. 7, Section 5, paragraph 6, I think that referring to EFda as a translation-defective RT inhibitor is too imprecise – it is not the inhibitor itself that is defective. I would describe it as an inhibitor that interferes with substrate translocation. Translocation is a property of RT and its substrate, not of the inhibitor.

p. 7, Section 5, paragraph 6, correct 2-fluro to 2-fluoro 

p. 7, Section 5, paragraph 6, last sentence: Change, "that provide" to "providing" 

p. 7, Section 6, correct first sentence to read: The discovery of nevirapine and TIBO inhibitors provided the foundation for further NNRTI discoveries [88, 89]. I had multiple problems with sentence 3 of this paragraph. First of all, I am not sure how selective the NNRTI binding pocket is. It clearly accommodates many different structures. I have previously read a review article mentioning that the NNRTI site is extremely promiscuous and binds many things – although perhaps many of these ligands lack high affinity or specificity. Unfortunately, I can't recall the reference, but perhaps the current authors may want to find this. Second, the sentence is ambiguous, since the phrase "designed with significantly low toxicity" sounds as if it is the design work itself that is toxic. Third, what is the difference between "significantly low toxicity" and "low toxicity"? As a final issue, the differential toxicity between NNRTI and NRTI results mostly from the fact that nucleosides are important to cellular function so that to varying extents NRTIs will compete with and interfere with these functions, while NNRTIs will only be toxic due to random, unanticipated interactions. This distinction is not clearly made. This entire paragraph needs to be revised.   

Pages 9-11.  It seems that Sections 7, 8, 9 and 10 should not be separate sections.  Section 7 is only two sentences.  Perhaps sections 8, 9 and 10 should be 7a, 7b, and 7c?  The authors should check for the preferred journal format. In general, I think that these sections will be of greatest interest to most RT researchers, who may only be familiar with well-described NRTIs and NNRTIs.  

General issues: 

            Since this paper emphasizes structure insights, it would be useful for the authors to more consistently include PDB accession codes.  Some are included, but others, for example, Figure 4 has not PDB ID for the rilpivirine structure; Figure 5 has no PDB IDs for the beta-thujaplicinol structure for the structure of the INDOPY-1 complex.  

Reviewer 2 Report

Singh and colleague have undertaken a high impact topic in retrovirology. The review is well organized but I have some observations  which the authors should address in order to make the manuscript suitable for publication.

1)The author must revise the manuscript  and check whether all the acronymous have been properly expressed in full.

2) I think that since the authors reproduce several figures it is not sufficient just to  indicate the font from which they reproduce as matter of fact that some panels in composite figures are identical to the original published. For example panel B in figure 4 is identical to the font. Therefore the authors should ask the permission to the original authors for all the figure reproductions. Alternatively, they have to change the figures to a more personal version.  In particular the panel B in figure 4 is not very clear neither in the original manuscript. Therefore the authors should reorganized the panel in more clear version. The “torsion” is not  very  evident. In addition I  suggest the authors to add some more detail in the section  describing a more recent approach in designing RT inhibitors.: The “P-pocket”.
